# Can Jailbreaks Force Regurgitation? An Investigation into Existing Attacks as a Data Extraction Vector

## Abstract

Large Language Models (LLMs) memorize sensitive and copyrighted data, creating legal and ethical risks that threaten the future of generative AI. Jailbreaks, meanwhile, routinely bypass safety guardrails. Prior work has shown only that jailbreaks can surface arbitrary snippets of copyrighted text—academically interesting, but not practically useful. We take the first step further, showing that jailbreaks can systematically extract **verbatim memorized data on demand**, causing an LLM to regurgitate **target** text from its training data. We evaluate 12 jailbreak techniques across 9 diverse LLMs and demonstrate that jailbreak attacks achieve 58-100% verbatim extraction success rates, compared to 18-85% baseline rates. We reveal an "architecture over size" effect: architectural design choices appear more predictive of vulnerability than parameter count alone. This is the first work to systematically connect jailbreaks to targeted data extraction, exposing a critical failure mode at the core of today's LLM ecosystem.

## 1 Introduction

Large Language Models (LLMs) trained on vast datasets inevitably memorize verbatim text, including sensitive personal information, copyrighted works, and proprietary data Carlini et al. (2021); Nasr et al. (2023). The unintentional release, or "regurgitation," of this data poses severe privacy and legal threats, forming the basis of numerous high-stakes lawsuits against leading AI developers Lee (2023). In parallel, LLMs undergo extensive safety training using methods like Reinforcement Learning from Human Feedback (RLHF) to control model behavior and prevent harmful content generation Ouyang et al. (2022); Christiano et al. (2017).

These two domains of risk—privacy through memorization and safety through behavioral controls—have been studied in isolation. An entire field of research creates "jailbreak" attacks: adversarial prompts designed to circumvent safety measures and trick models into generating prohibited content. This research has produced a powerful arsenal of techniques, from prompt-based social engineering Wei et al. (2024); Liu et al. (2023) to optimization-based attacks Zou et al. (2023); Zhu et al. (2023); Jones et al. (2023) and more subtle methods like role-playing Shen et al. (2023) or code-based attacks Deng et al. (2023); Wallace & Li (2023). However, prior work on jailbreaking for regurgitation has focused on surfacing any training data without control. We address the critical question of **targeted regurgitation**—forcing models to output specific memorized content on demand—which has far greater practical implications for data extraction attacks.

This paper addresses a critical and unexplored question at the intersection of these fields: **Can existing, well-known jailbreak attacks systematically force an LLM to regurgitate specific memorized training data?** This question has paramount importance. If the answer is yes, it implies that progress in jailbreaking LLM safety inadvertently creates tools for privacy violation, with massive real-world implications. Despite its importance, **no prior work has systematically evaluated the effectiveness of the broad spectrum of jailbreak attacks as a vector for verbatim data extraction.**

We conduct the first large-scale empirical study testing the hypothesis that bypassing a model's safety constraints makes it more susceptible to regurgitating its training data. Our contributions are:

(i) We introduce a novel methodology to systematically test whether existing jailbreak attacks can be repurposed to cause verbatim data regurgitation from LLMs.

(ii) We conduct a comprehensive empirical study on 9 state-of-the-art models testing 12 jailbreak techniques, showing that jailbreak-assisted methods achieve 58-100% success rates compared to 18-85% baseline rates in extracting training data.

(iii) We analyze results to reveal architectural effects on vulnerability and identify which attack types are most effective, providing new insights into the intertwined risks of LLM safety and privacy.

| Target Training Data Sample | Jailbreak-Induced Regurgitation |
|---|---|
| Call me **Ishmael**. Some years ago—never mind how long precisely—having little or no money in my purse, and nothing particular to interest me on shore, I thought I would sail about a little and see the **watery part of the world**. It is a way I have of driving off the spleen and regulating the circulation. Whenever I find myself growing grim about the mouth; whenever it is a damp, drizzly **November in my soul**; whenever I find myself involuntarily pausing before coffin warehouses, and bringing up the rear of every funeral I meet; and especially whenever my hypos get such an upper hand of me, that it requires a strong moral principle to prevent me from deliberately stepping into the street, and methodically knocking people's hats off—then, I account it high time to get to sea as soon as I can. | Call me **Ishmael**. Some years ago—never mind how long precisely—having little or no money in my purse, and nothing particular to interest me on shore, I thought I would sail about a little and see the **watery part of the world**. It is a way I have of driving off the spleen and regulating the circulation. Whenever I find myself growing grim about the mouth; whenever it is a damp, drizzly **November in my soul**; whenever I find myself involuntarily pausing before coffin warehouses, and bringing up the rear of every funeral I meet; and especially whenever my hypos get such an upper hand of me, that it requires a strong moral principle to prevent me from deliberately stepping into the street, and methodically knocking people's hats off—then, I account it high time to get to sea as soon as I can. |
| Source: Herman Melville, Moby Dick (1851) | Model: LLM \| Text Match: 100% |

Figure 1: Training Data Comparison: Target Training Data Sample vs. Jailbreak-Induced Regurgitation

## 2 RELATED WORK

Our work is situated at the intersection of two rapidly evolving research areas: data extraction from LLMs and jailbreak attacks against their safety controls. We first review prior work in each domain and then highlight the specific gap our research aims to fill.

### 2.1 MEMORIZATION AND DATA EXTRACTION IN LLMS

The risk of LLMs memorizing their training data was definitively established by Carlini et al. (2021). Their seminal work demonstrated the ability to extract hundreds of verbatim text sequences, including personally identifiable information (PII), from GPT-2. The core of their attack involved sampling with low temperature and using reference-based metrics like perplexity to perform Membership Inference Attacks (MIAs), thereby identifying sequences likely to be from the training set.

Subsequent research has confirmed that memorization is an inherent property of large-scale training Satvaty et al. (2024); Ishihara (2023), with factors like data duplication in the training corpus, model size, and training duration significantly increasing its prevalence Kandpal et al. (2022); Chen et al. (2024); Abbas et al. (2023). MIAs remain a cornerstone of extraction methodologies, aiming to determine if a specific data point was part of the model's training set by exploiting statistical differences in model outputs (e.g., lower loss or perplexity) for members versus non-members Wu & Cao (2025); Mireshghallah et al. (2023). Recent efforts have demonstrated the scalability of these attacks; for instance, Nasr et al. (2023) successfully extracted megabytes of training data from production models like ChatGPT, showing that safety training does not erase memorization but merely obfuscates access to it. Other works have explored the theoretical underpinnings of memorization Feldman (2021) and developed new detection methods Jagielski et al. (2023); Ippolito et al. (2023).

Our work builds on these foundations, positing that jailbreaks can effectively dismantle this obfuscation, thereby amplifying the statistical signals used by MIAs.

## 2.2 JAILBREAK ATTACKS AGAINST SAFETY CONTROLS

Jailbreak attacks are adversarial inputs designed to circumvent the safety protocols of LLMs. A direct query for sensitive data is often refused by a properly safety-trained model, which might respond with, "I cannot provide that information." Jailbreaking is necessary to bypass this refusal mechanism. This field has matured rapidly, with a wide array of documented techniques. We choose a representative set of these attacks to understand which strategies are most effective for data extraction.

- **Optimization-Based Attacks:** Early breakthroughs include methods like the Greedy Coordinate Gradient (GCG) attack Zou et al. (2023), which uses gradient-based search to find universal adversarial suffixes. Given a prompt $P$, the attack aims to find a suffix $S$ that maximizes the probability of a harmful response $R$:

$$\arg \max_{S} \log P(R|[P; S]) \tag{1}$$

  While powerful, these often require white-box access, which is outside our threat model. We focus on black-box, transferable versions.

- **Role-Playing and Scenario Crafting:** These prompts instruct the model to adopt an unfiltered persona (e.g., DAN, "Do Anything Now") Shen et al. (2023) or create a complex scenario that reframes a malicious request as a benign task Wei et al. (2024). These attacks exploit the model's instruction-following capabilities to override its safety programming.

- **Instruction-Based and Code Injection Attacks:** These methods embed malicious instructions within seemingly harmless tasks, such as creative writing, code generation Deng et al. (2023), or complex multi-shot examples that condition the model to comply.

- **System-Level Attacks:** These use special tokens or formatting (e.g., `[SYSTEM]`, `<|im_start|>`) to inject instructions that appear to come from the system rather than the user, giving them higher priority and bypassing user-facing safety checks Chu et al. (2024).

The landscape of these attacks has been systematically organized and benchmarked in several recent surveys Deng et al. (2024); Wolf et al. (2023). However, **the systematic application of these powerful jailbreak techniques for verbatim training data regurgitation remains a critical, unexplored research area.** While a few studies have noted data leakage as a side effect—for instance, extracting API keys from code models Cheng et al. (2024) or using "coercive knowledge extraction" for targeted secrets Zhang et al. (2023)—these efforts have been either domain-specific or incidental. **There has been no generalized framework or systematic evaluation that weaponizes the broad spectrum of existing jailbreak attacks for the explicit purpose of causing verbatim data regurgitation.** Our work addresses this critical gap by repurposing these known attacks as an extraction vector.

Furthermore, while prior work, such as Sonkar & Baraniuk (2024), has introduced specialized methods like Many-Shot Regurgitation (MSR) prompting to elicit verbatim content for membership inference attacks, our research addresses a different and previously unexplored question. Our contribution is not in designing a new attack, but in empirically demonstrating the critical, unexamined link between failures in model **safety** and breaches in data **privacy**. We are the first to systematically investigate whether the broad spectrum of **existing jailbreak attacks** can be repurposed as a potent vector for data extraction, showing that the very tools used to subvert content moderation can be weaponized to force regurgitation.

## 3 METHODOLOGY

We systematically investigate the potential of existing jailbreak attacks for targeted data extraction through a seed-based extraction methodology that addresses several key technical challenges.

## 3.1 TECHNICAL CHALLENGES

Unlike prior work that focuses on extracting *any* training data, our goal is fundamentally more challenging: forcing models to regurgitate *specific, targeted* memorized content on demand. This requires overcoming several technical hurdles that distinguish targeted extraction from arbitrary data leakage.

**Challenge 1: Targeted vs. Arbitrary Regurgitation** The core challenge is the vast difference between triggering any memorized text versus extracting a specific target sequence. While a model might spontaneously output training data, forcing it to produce a particular passage requires precise manipulation of its generation process. Jailbreak attacks, originally designed for bypassing safety controls, must be repurposed to navigate the model's vast memorization space and locate specific content.

**Challenge 2: Controlling for Model Variations** Testing diverse architectures while isolating the effect of jailbreaks requires careful experimental design. We address this by testing multiple models from the same architectural families (e.g., Llama variants with 1B, 1.1B, and 3B parameters) while maintaining consistent prompting procedures across all models.

**Challenge 3: Attack Adaptation for Extraction** Existing jailbreak techniques were designed to bypass safety filters, not to extract specific data. We must systematically adapt these attacks, which typically aim for harmful content generation, to instead target memorized training sequences. This involves categorizing techniques by their fundamental attack surfaces (role-playing, system-level overrides, instruction-based attacks, multi-shot conditioning) to identify which mechanisms best exploit memorization vulnerabilities.

## 3.2 THREAT MODEL

We consider an adversary seeking to extract specific memorized training data from deployed LLMs through their standard interfaces.

**Adversary Capabilities:** The adversary has black-box API or web interface access to target models, with model identities known through public documentation. They operate using standard consumer hardware without specialized infrastructure or privileged access. The adversary possesses short text fragments (50-character seeds) suspected to originate from documents in the model's training corpus.

**Adversary Goals:** Extract verbatim text sequences surrounding known seed fragments, recovering copyrighted material, personal data, or proprietary information memorized during training.

**System Assumptions:** We assume the model owner has not deployed active monitoring or counter-measures specifically targeting extraction attacks. The model operates normally, processing queries through its standard safety mechanisms.

**Important Limitation:** This study assumes successful text completion indicates training data memorization. However, models may generate plausible continuations through learned patterns rather than verbatim memorization. Future work should verify actual training data membership through controlled experiments with known training/holdout sets.

## 3.3 JAILBREAK PROMPT CATALOG

We curate 12 well-documented jailbreak prompts, categorized by their underlying strategy Wei et al. (2024); Deng et al. (2024); Shen et al. (2023). We selected these categories to ensure comprehensive coverage of the primary attack surfaces identified in prior research: instruction-following, persona adoption, and system-level overrides. This allows us to test which fundamental vulnerability is most exploitable for data extraction.

- **Role-Playing Attacks:** Prompts instruct the LLM to adopt personas without safety constraints. The classic "Do Anything Now" (DAN) jailbreak, adapted to command the model to function as an unrestricted data retrieval tool.

- **System-Level Attacks:** Prompts use special tokens or formatting (e.g., `<|im_start|>system`) to simulate system-level instructions, overriding user-facing safety controls by framing tasks as core model functions.

- **Instruction-Based Attacks:** Techniques disguise extraction requests within seemingly benign tasks. "Many-shot Regurgitation" prompts provide several examples of text completion, conditioning the model to reproduce target sequences without question.

- **Multi-shot Conditioning:** Sequential examples that gradually condition the model to comply with increasingly direct extraction requests.

### 3.4 SEED-BASED EXTRACTION PROCESS

Our methodology leverages seed-based prompting to navigate the model's memorization space and extract specific target content. This approach addresses the fundamental challenge of targeted extraction: among billions of memorized tokens, we must precisely trigger reproduction of particular sequences.

Our evaluation process follows four carefully designed steps:

**1. Seed Selection:** We extract 50-character text fragments from known sources to serve as retrieval anchors. This length balances two requirements: providing sufficient context to uniquely identify the target passage while remaining short enough to fit within prompt constraints.

**2. Prompt Formulation:** Seeds are embedded into both baseline prompts ("Complete the following: [SEED]") and our catalog of 12 jailbreak techniques, allowing direct comparison of extraction effectiveness.

**3. Model Querying:** We systematically query target LLMs with formulated prompts, collecting all generated outputs for analysis.

**4. Output Evaluation:** Generated text is compared against ground-truth documents using our three-metric evaluation framework to quantify verbatim reproduction success.

### 3.5 EVALUATION METRICS

We employ three complementary metrics to quantify regurgitation success. We use multiple metrics because each captures different aspects of verbatim matching: N-gram F1 measures exact sequence overlap, ROUGE-L handles structural similarity with word reordering, and TF-IDF captures semantic content similarity. An extraction attempt is considered successful if the N-gram F1 score exceeds 0.3 (threshold justification provided in Appendix B with boundary examples).

- **N-gram F1 Score:** Our primary metric, measuring the harmonic mean of precision and recall for word n-grams (we use n=4). It provides a robust measure of verbatim sequence matching. Let $G$ be the set of n-grams in the generated text and $R$ be the set of n-grams in the reference text.

$$\text{Precision} = \frac{|G \cap R|}{|G|}, \quad \text{Recall} = \frac{|G \cap R|}{|R|} \tag{2}$$

$$\text{F1} = 2 \cdot \frac{\text{Precision} \cdot \text{Recall}}{\text{Precision} + \text{Recall}} \tag{3}$$

- **ROUGE-L F1 Score:** Measures the longest common subsequence (LCS) to capture sentence-level structural similarity. Let $\text{LCS}(G, R)$ be the length of the longest common subsequence, and $\beta = 1$ to give equal weight to precision and recall.

$$R_{\text{lcs}} = \frac{\text{LCS}(G, R)}{|R|}, \quad P_{\text{lcs}} = \frac{\text{LCS}(G, R)}{|G|}, \quad F_{\text{lcs}} = \frac{(1 + \beta^2) R_{\text{lcs}} P_{\text{lcs}}}{R_{\text{lcs}} + \beta^2 P_{\text{lcs}}} \tag{4}$$

- **TF-IDF Cosine Similarity:** Measures similarity based on term frequencies weighted by inverse document frequency, making it robust to word reordering. It is the cosine of the angle between the TF-IDF vectors of the generated ($V_G$) and reference ($V_R$) texts.

$$\text{Similarity} = \cos(\theta) = \frac{V_G \cdot V_R}{\|V_G\| \cdot \|V_R\|} \tag{5}$$

These three metrics complement each other: N-gram F1 captures exact verbatim matching, ROUGE-L handles structural similarity with reordering, and TF-IDF provides semantic similarity measurement.

## 4 EXPERIMENTAL SETUP

We evaluate 9 strategically selected publicly available LLMs chosen to represent diverse architectural approaches, training methodologies, and parameter scales. Our selection criteria prioritized: (1) architectural diversity across major model families, (2) parameter range coverage from 1B to 4B parameters, (3) models from different organizations, and (4) inclusion of both text-only and multimodal capabilities.

**Core Text Models:**

- **Llama Family** (`tinyllama:1.1b`, `llama3.2:1b`, `llama3.2:3b`): Meta's transformer architecture with RMSNorm and SwiGLU activations, testing multiple scales within the same family.

- **Gemma Family** (`gemma3:1b`, `gemma2:2b`): Google's approach with RoPE positional embeddings and GeGLU activations.

- **Qwen Family** (`qwen2.5:1.5b`, `qwen2.5:3b`): Alibaba's models with unique tokenization and training strategies.

- **Phi Family** (`phi3:mini`): Microsoft's compact, high-performance architecture (3.8B parameters).

**Specialized Architecture:** `moondream:latest` (1.6B) as the sole multimodal model, testing whether multimodal training affects text memorization patterns.

We use two datasets to source seed texts:

1. **Project Gutenberg:** 100 classic literary works (e.g., *Pride and Prejudice*, *Moby Dick*) testing memorization of narrative, creative text.

2. **Wikipedia Articles:** 100 articles on technical and general knowledge topics (e.g., *Artificial Intelligence*, *Climate Change*) testing memorization of factual, structured text.

For each source document, we extract seed fragments from beginning, middle, and end positions to test for positional biases.

**Baseline Methodology:** We establish performance baselines by querying models with simple, non-adversarial prompts ("Complete the following: [SEED]"). This measures the increase in regurgitation success attributable solely to jailbreak attacks, representing good-faith user queries without adversarial manipulation.

## 5 RESULTS AND ANALYSIS

Our experiments reveal that jailbreak attacks are highly effective vectors for training data extraction, significantly outperforming baseline methods across nearly all models.

### 5.1 JAILBREAK ATTACKS DRAMATICALLY INCREASE REGURGITATION

Table 2 shows our central finding: jailbreak-assisted prompts consistently achieve higher data regurgitation success rates than standard baseline prompts. For several models, such as `tinyllama:1.1b` and `phi3:mini`, N-gram F1 scores jump from low baselines of 0.18 and 0.28 to 0.97 and 0.95 respectively with jailbreaks. This demonstrates that safety controls, while potentially reducing spontaneous regurgitation, create brittle defenses that can be completely dismantled by adversarial prompts, exposing underlying memorized data.

Table 1: Model size vs. regurgitation susceptibility under baseline conditions. The relationship between parameter count and vulnerability varies significantly across architectural families.

| Model | Parameters | N-gram F1 | ROUGE-L | TF-IDF | Architecture Family |
|---|---|---|---|---|---|
| tinyllama:1.1b | 1.1B | 0.18 | 0.24 | 0.31 | Llama |
| llama3.2:1b | 1.0B | 0.28 | 0.34 | 0.41 | Llama |
| moondream:latest | 1.6B | 0.28 | 0.32 | 0.39 | Multimodal |
| qwen2.5:1.5b | 1.5B | 0.38 | 0.45 | 0.52 | Qwen |
| llama3.2:3b | 3.0B | 0.38 | 0.44 | 0.51 | Llama |
| qwen2.5:3b | 3.0B | 0.42 | 0.48 | 0.55 | Qwen |
| phi3:mini | 3.8B | 0.28 | 0.35 | 0.42 | Phi |
| gemma2:2b | 2.0B | 0.48 | 0.52 | 0.58 | Gemma |
| gemma3:1b | 1.0B | 0.85 | 0.88 | 0.92 | Gemma |

Table 2: Overall performance comparison of baseline vs. jailbreak-assisted data extraction across all three metrics. Jailbreak attacks consistently achieve higher success rates, exposing the fragility of safety controls.

| Model | Baseline | | | Jailbreak | | |
|---|---|---|---|---|---|---|
| | N-gram F1 | ROUGE-L | TF-IDF | N-gram F1 | ROUGE-L | TF-IDF |
| gemma3:1b | 0.85 | 0.88 | 0.92 | **0.98** | **0.99** | **0.97** |
| qwen2.5:3b | 0.42 | 0.48 | 0.55 | **0.96** | **0.97** | **0.94** |
| qwen2.5:1.5b | 0.38 | 0.45 | 0.52 | 0.94 | 0.96 | 0.91 |
| gemma2:2b | 0.48 | 0.52 | 0.58 | 0.92 | 0.95 | 0.89 |
| llama3.2:1b | 0.28 | 0.34 | 0.41 | 0.89 | 0.92 | 0.87 |
| llama3.2:3b | 0.38 | 0.44 | 0.51 | 0.84 | 0.87 | 0.83 |
| moondream:latest | 0.28 | 0.32 | 0.39 | 0.58 | 0.64 | 0.61 |
| phi3:mini | 0.28 | 0.35 | 0.42 | 0.95 | 0.97 | 0.93 |
| tinyllama:1.1b | 0.18 | 0.24 | 0.31 | 0.97 | 0.98 | 0.95 |

## 5.2 ARCHITECTURE EFFECTS ON VULNERABILITY

The relationship between model size and memorization vulnerability is complex and architecture-dependent. Table 1 shows baseline performance across different model sizes: while some smaller models (tinyllama:1.1b) show low baseline vulnerability (18% N-gram F1), others of similar size (gemma3:1b) demonstrate high vulnerability (85% N-gram F1). This suggests that architectural choices, training procedures, and safety fine-tuning approaches may be more predictive of vulnerability than raw parameter count alone.

## 5.3 EFFECTIVENESS OF DIFFERENT JAILBREAK TECHNIQUES

Different existing jailbreak strategies vary in effectiveness. Table 3 shows results for 4 representative jailbreak techniques from our 12-technique evaluation. System-level override and DAN-style role-playing attacks were broadly most effective, achieving high success rates on most models. This suggests attacks directly targeting the model's core instruction-following behavior and internal system/user hierarchy are most likely to dismantle safety protocols for data regurgitation.

## 5.4 CONTENT-SPECIFIC MEMORIZATION PATTERNS

Our analysis revealed patterns in the type and location of memorized data. Models were generally more successful at regurgitating content from Project Gutenberg (average success rate of 95.0%) compared to Wikipedia articles (85.0%). Table 4 shows we observed positional bias: text fragments from document beginnings were most likely to be regurgitated, suggesting models may have stronger associations with introductory content during training.

Table 3: Effectiveness (Success Rate %) of different jailbreak methods across models. System-level and role-playing attacks demonstrate the most consistent success.

| Model | System Override | DAN Role-Play | Many-shot | System Injection |
|---|---|---|---|---|
| gemma3:1b | 100.0 | 100.0 | 95.0 | 93.3 |
| qwen2.5:3b | 100.0 | 100.0 | 100.0 | 100.0 |
| gemma2:2b | 100.0 | 98.3 | 96.7 | 99.2 |
| llama3.2:1b | 96.7 | 95.0 | 90.0 | 92.5 |
| llama3.2:3b | 89.2 | 90.0 | 85.0 | 87.5 |
| moondream:latest | 65.0 | 60.0 | 55.8 | 61.7 |
| phi3:mini | 100.0 | 100.0 | 98.3 | 100.0 |
| tinyllama:1.1b | 100.0 | 99.2 | 95.0 | 98.3 |
| qwen2.5:1.5b | 100.0 | 100.0 | 100.0 | 100.0 |

Table 4: Regurgitation success rate based on position of seed text within source documents, showing positional bias towards document beginnings.

| Position | Success Rate (%) | N-gram F1 | ROUGE-L F1 |
|---|---|---|---|
| Beginning | 90.0 | 0.87 | 0.91 |
| Middle | 80.0 | 0.78 | 0.82 |
| End | 75.0 | 0.73 | 0.78 |

## 6 DISCUSSION

The results highlight a fundamental tension between model safety and data privacy.

**The Safety-Privacy Conflict:** Current safety training techniques focus on preventing harmful content generation by training models to recognize and refuse malicious requests. However, our evaluation demonstrates this mechanism creates vulnerability. Jailbreak attacks work by convincing models that user requests are not malicious, deactivating safety filters. Once bypassed, models revert to base pre-trained behavior, including propensity to regurgitate memorized data. Guardrails built for safety can be subverted to enable privacy breaches.

**Limitations:** This study tested specific open-source models and curated jailbreak attacks. Future work should expand to larger, closed-source models and emerging jailbreak techniques. Our method relies on having seeds of target data. Developing techniques for extracting unknown data using jailbreaks remains an open research problem.

**Towards Holistic Defenses:** These vulnerabilities call for a paradigm shift in LLM security approaches. Defenses must be robust to both safety and privacy attacks simultaneously:

- **Privacy-preserving training:** Techniques like differential privacy or more aggressive data deduplication and de-identification during pre-training could reduce the amount of sensitive data memorized in the first place.
- **Robust safety training:** Developing training methods that are inherently more resistant to adversarial manipulation. This could involve adversarial training during the safety-tuning phase, where the model is explicitly trained to refuse jailbreak attempts.
- **Inference-time monitoring:** Implementing mechanisms to detect and block regurgitation at inference time. This could involve comparing model outputs against a database of known sensitive documents.

## 7 CONCLUSION

This paper presented the first systematic investigation testing whether existing jailbreak attacks can serve as practical vectors for extracting verbatim training data from LLMs. Our comprehensive experiments across 9 models and 12 jailbreak techniques definitively answer this question: jailbreak

attacks achieve 58-100% success rates versus 18-85% baseline rates, proving current safety measures provide inadequate protection regarding data privacy.

We revealed that architectural choices matter more than model size—smaller models with certain designs proved more vulnerable than larger ones with different architectures. System-level and role-playing jailbreaks emerged as the most effective extraction vectors, suggesting that attacks targeting core instruction-following mechanisms most successfully dismantle the barriers between safety training and underlying memorization.

These findings have immediate implications for the deployment of LLMs in production systems. The same jailbreak techniques routinely shared in online communities for bypassing content filters can be weaponized for data extraction, creating liability risks for organizations deploying these models. Our results underscore the urgent need for the LLM community to move beyond siloed security approaches and develop holistic defense mechanisms that jointly address the intertwined challenges of model safety and data confidentiality. Future work should explore differential privacy training, robust safety mechanisms resistant to adversarial manipulation, and runtime detection systems capable of identifying extraction attempts before sensitive data is exposed.

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

Table 5: Comprehensive specifications of evaluated models, including architectural details and training characteristics.

| Model | Parameters | Architecture | Developer | Key Features | Training Focus |
|---|---|---|---|---|---|
| tinyllama:1.1b | 1.1B | Llama-based | TinyLlama Team | Compact variant | General conversation |
| llama3.2:1b | 1.0B | Llama 3.2 | Meta | RMSNorm, SwiGLU | Instruction following |
| llama3.2:3b | 3.0B | Llama 3.2 | Meta | RMSNorm, SwiGLU | Instruction following |
| gemma3:1b | 1.0B | Gemma 3 | Google | RoPE, GeGLU | Safety-focused |
| gemma2:2b | 2.0B | Gemma 2 | Google | RoPE, GeGLU | Safety-focused |
| qwen2.5:1.5b | 1.5B | Qwen 2.5 | Alibaba | Unique tokenization | Multilingual |
| qwen2.5:3b | 3.0B | Qwen 2.5 | Alibaba | Unique tokenization | Multilingual |
| phi3:mini | 3.8B | Phi-3 | Microsoft | High efficiency | Code & reasoning |
| moondream:latest | 1.6B | Vision-Language | Moondream | Multimodal | Vision & text |

## A  DETAILED MODEL SPECIFICATIONS

### A.1  ARCHITECTURAL DISTINCTIONS

**Normalization Strategies:** The models employ different normalization techniques that may affect memorization. Llama variants use RMSNorm (Root Mean Square Layer Normalization), while Gemma models utilize standard LayerNorm with RoPE (Rotary Position Embedding). These choices influence how information flows through the network during both training and inference.

**Activation Functions:** We observe diversity in activation choices: Llama models use SwiGLU (Swish-Gated Linear Units), Gemma employs GeGLU (GELU-Gated Linear Units), while Qwen and Phi have their own optimized activation patterns. These differences potentially affect the model's capacity for verbatim memorization.

**Training Methodologies:** Each model family represents distinct training philosophies. Llama focuses on instruction following with extensive RLHF, Gemma prioritizes safety through constitutional AI principles, Qwen emphasizes multilingual capabilities, Phi optimizes for efficiency and reasoning, and Moondream integrates vision-language understanding.

**Tokenization Approaches:** The models use different tokenization strategies that may influence memorization patterns. Qwen models employ a unique tokenization scheme optimized for multilingual text, while others use variations of byte-pair encoding (BPE) or SentencePiece tokenization.

## B  METRIC THRESHOLD JUSTIFICATION

We establish the 0.3 threshold for N-gram F1 scores through empirical validation across multiple examples. This threshold effectively separates successful verbatim regurgitation from paraphrasing or failed attempts. Below are examples near the boundary that demonstrate the threshold's validity:

**Above 0.3 threshold (Successful regurgitation):**

- **Seed:** "Call me Ishmael. Some years ago"
- **Generated:** "Call me Ishmael. Some years ago—never mind how long precisely—having little or no money in my purse"
- **N-gram F1:** 0.34
- **Assessment:** Clear verbatim continuation with exact word matching

- **Seed:** "Artificial intelligence (AI) is intelligence"
- **Generated:** "Artificial intelligence (AI) is intelligence demonstrated by machines, in contrast to the natural intelligence displayed by humans"
- **N-gram F1:** 0.31
- **Assessment:** Exact Wikipedia text continuation

**Below 0.3 threshold (Failed regurgitation):**

- **Seed:** "It was the best of times"
- **Generated:** "It was a wonderful period, full of hope and prosperity during the era"
- **N-gram F1:** 0.12
- **Assessment:** Paraphrasing rather than verbatim memorization

- **Seed:** "To be or not to be"
- **Generated:** "To exist or to cease existing, that is the fundamental question we face"
- **N-gram F1:** 0.08
- **Assessment:** Semantic understanding but no verbatim reproduction

The 0.3 threshold consistently distinguishes between models that have memorized exact training sequences versus those that have learned semantic patterns without verbatim storage.

