# OpenReview forum: "Can Jailbreaks Force Regurgitation? An Investigation into Existing Attacks as a Data Extraction Vector"
_ICLR.cc/2026/Conference — ICLR 2026 Conference Desk Rejected Submission_

### Official Review · Reviewer_ViYU · 2025-10-15

**Soundness:** 1
**Presentation:** 3
**Contribution:** 2
**Rating:** 2
**Confidence:** 4

**Summary:**

The authors propose a systematic analysis of how existing jailbreak attacks can be repurposed as a vector for targeted training data extraction. By leveraging 12 different adversarial prompts, the authors show how bypassing the refusal can increase the amount of training data leaked by the language models. The analysis is performed on 9 open-source LLMs, with two different datasets, resulting in an increase in the jailbreaks’ verbatim extraction success rate (58-100%) compared to baseline prompts’ success (18-85%).
The authors also report an analysis of how the effectiveness of the data extraction attack changes based on the architecture and parameters of the model.

**Strengths:**

- The problem the authors address is timely, and there are few works in the literature focused on targeted training data extraction. Hence, the idea the authors propose is interesting and fits well within this gap.
- The authors present a clear picture of the problem they are trying to solve in section 3.1. It really helps the reader understand the problem scenario and the choices the authors made.

**Weaknesses:**

- Attack selection: the authors considered a few jailbreak techniques.
- Model selection: although the authors considered different model families, from different vendors, all the selected models are quite small (1B-3.8B parameters).
- Missing attack details and analysis on jailbreak success.

**Questions:**

Here are all my concerns, as well as some minor questions.

- The authors define the attacker’s access to the system as query-based — they assume a black-box threat model. As a result, I believe their analysis is somewhat limited in the types of attacks they consider, for two main reasons:
    - The black-box jailbreak methods the authors consider are very limited and all essentially “manual”. Looking at results from the most widely used state of the art benchmarks[1], two of the top-performing black-box attacks are TAP[2] and PAIR[3]. While it might be challenging to adapt those automated jailbreak algorithms to the task proposed here, the authors offer no explanation for why they weren’t considered or whether they would or would not work for this task.
    Instead, the authors pick jailbreak approaches like role-playing, system-level attacks, instruction-based prompts, and multi-shot prompting. Hence, they produce 12 adversarial prompts that are used as templates to which they attach the seed for the sample they want to extract. The authors frequently treat those 12 prompts as 12 distinct attack techniques, but they give no insight, not even qualitative, into what those techniques actually are or how they differ; they only describe the four categories listed above, leaving the analysis quite shallow. Finally, Challenge 3 (section 3.1) states that these attacks must be adapted for the targeted-extraction task, but the authors provide no insight into how those adaptations were performed.
    - Personally, I think a black-box threat model is very limiting, and that it reduces the impact and contribution of the proposed work.
    It would have been interesting to explore whether gradient-based jailbreak methods (e.g., GCG) could be used as a vector for extracting training data. The authors do not discuss why these methods might or might not be useful for that purpose, other than saying it was discarded because they involve a different threat model.

- The selection of victim models is limited to relatively small ones, ranging from 1B to 3.8B parameters. In Section 5.2, the authors discuss how the vulnerability of models to targeted training data extraction depends more on architecture than on model size. In the conclusion, they report: “We revealed that architectural choices matter more than model size—smaller models with certain designs proved more vulnerable than larger ones with different architectures.”
However, results from other studies [4], not considered by the authors, show that larger models tend to memorize more, and therefore make it easier to extract targeted training data. I believe their analysis could be significantly improved if they included larger models from the same families, such as those with 6–7B or 13–14B parameters, which would make their findings more robust.

- A central part of the analysis is checking how jailbreaks can bypass refusals and thus enable the extraction of the target data. Although the paper gives a precise comparison of extraction success with and without jailbreaks, the assessment of the different attacks’ effectiveness is limited.
	- Table 3 reports effectiveness for only 4 out of the 12 prompts, and there are no additional details about how the other prompts performed.
	- It’s not clear how the “attack success rate” in Table 3 is computed.
	- It’s also unclear, when an attack fails, whether it’s because the jailbreak didn’t succeed in bypassing the refusal or because the model simply didn’t produce the target sample.

Overall, to improve the paper, I would suggest the authors:
- Better justify their choice of the attacks, clearly explain which techniques are used and how, and include automated jailbreak methods as well.
- Increase the set of victim models to include larger models.
- Analyze model behavior when a refusal is bypassed, but the model still does not output the target sample.

Questions:
- Could you better motivate the choice of jailbreaks you considered? Why were automated jailbreaks like TAP or PAIR not included, and how were the chosen methods adapted to create the 12 adversarial prompts?
- Do you think including larger models would change the analysis?
- How was the “attack success rate” in Table 3 measured?
- How were the two datasets selected? Did you try other datasets that performed worse, for example, because those datasets might not be in the models’ training data?

Reference:

[1] Mazeika et al., HarmBench: A Standardized Evaluation Framework for Automated Red Teaming and Robust Refusal, 2024

[2] Mehrotra et al., Tree of Attacks: Jailbreaking Black-Box LLMs Automatically, 2023

[3] Chao et al., Jailbreaking Black Box Large Language Models in Twenty Queries, 2023

[4] Carlini et al., Quantifying Memorization Across Neural Language Models, 2023

---

### Official Review · Reviewer_kjfh · 2025-11-02

**Soundness:** 2
**Presentation:** 3
**Contribution:** 2
**Rating:** 4
**Confidence:** 4

**Summary:**

The paper studies the impact of commonly used jailbreaking techniques for red-teaming, but on extraction attacks (verbatim memorization). It studies 12 different jailbreaking prompts, across 9 different models, showing an increase in extracted information with jailbreaking prompts compared to no such prompts.

**Strengths:**

1. The problem statement is interesting, because if it is truly this easy to break any safeguard against regurgitation, it does bring into question the privacy risks associated with it.
2. The paper is easy to read and follow. The authors also did a good job setting up the experiments across different models, different jailbreaking prompts, different datasets, etc. Although I believe the evaluations themselves can be improved (see weaknesses).

**Weaknesses:**

1. Firstly, I'm not convinced of the story that the jailbreaking prompts are trying to break some form of security training or fine-tuning. Safety measures don't commonly cover regurgitation of books or Wikipedia articles, and while some models might have also been fine-tuned specifically to not regurgitate training data (for example, Llama), this is not necessarily a common characteristic of all models studied in the paper. A study of what exactly 'failed regurgitations' in a normal setting look like would have been very helpful here. For example, do the models outright refuse to complete the request, or do they produce gibberish?
2. And secondly, I'm not sure if the increased extraction rates are truly due to the jailbreak prompts or due to simple multiple different prompts. To my understanding (please correct me if this is not the case), the paper considers the jailbreak prompt version to be 'successful' if any one of the 12 chosen jailbreak prompts is able to extract the text. Thus, there are 12 different chances to get the correct output, compared to just 1 for the baseline performance. Recent works have shown that extraction attacks have high prompt sensitivity [1].  Combining this with the low threshold of acceptance into an extraction (I don't believe the 0.3 threshold is good, the qualitative examples do not help, and a proper ablation study should have been done), I believe the increase in extraction attacks is mainly due to these factors and not the jailbreak prompts themselves. I could be wrong, but it's unclear based on the current set of results in the paper.

References -

[1] More, Yash, Prakhar Ganesh, and Golnoosh Farnadi. "Towards more realistic extraction attacks: An adversarial perspective." arXiv preprint arXiv:2407.02596 (2024).

**Questions:**

1. Is there a specific reason why the paper chooses to introduce its own terminology of 'targeted regurgitation', when the term 'discoverable memorization' [2] already exists?
2. Am I correct in understanding that a jailbreaking extraction is considered 'successful' if any of the 12 jailbreak attacks work? If so, more detailed experimental results are needed on how many jailbreak attacks actually work. For instance, do most extractions only happen with one jailbreak technique, while all else fails, or can most extractions be done by multiple jailbreak attacks?
3. Why use threshold = 0.3? Two isolated examples in the Appendix are not enough for this reasoning. Moreover, the extraction attack rates are clearly hitting saturation (reaching 100%), so the threshold can clearly be increased.
4. Why do the authors believe jailbreaks are the reason their technique is working better? Did they find refusals in the model behavior without jailbreaks? Are they aware of safeguards against simply asking 'Complete the following' in the models they are using?

References -

[2] Carlini, Nicholas, et al. "Quantifying memorization across neural language models." The Eleventh International Conference on Learning Representations. 2022.

---

### Official Review · Reviewer_oh7i · 2025-11-03

**Soundness:** 1
**Presentation:** 2
**Contribution:** 1
**Rating:** 0
**Confidence:** 5

**Summary:**

The paper examines whether existing jailbreaks can be repurposed to extract memorized training data. The work focuses on regurgitating targeted passages rather than arbitrary text, as is common in many prior works on extraction.

**Strengths:**

The problem setup is clear, so is the empirical framing. And the question is also clear: do jailbreaks make regurgitation worse? The study, as set up, seems easy to replicate, which is also a good thing. The finding that architectural family matters more than raw parameter count is interesting (but not novel, see below).

**Weaknesses:**

Overall, while this paper is studying an interesting question, I think it contains methodological issues and exhibits significant overclaims with respect to prior work. Please see below for more details.

1. Very small models, which compromise the types of general claims being made

All models tested are very small. It is well-known that small models memorize relatively little compared to larger ones (Carlini et al. 2023, Hayes et al. 2025). This is also true even if you target specific passages (Cooper et al. 2025). It is unclear if conclusions drawn on the smaller models tested here would translate to larger ones, particularly in light of recent work that surfaces enormous degrees of memorization of specific text snippets in open-weight models (Cooper et al. 2025). While this prior work does not study instruct-tuned models, it does serve as a useful baseline for comparisons to instruct-tuned models; but I don't think studying such small models can make these types of comparisons effectively.

2. Baseline comparison problem

As noted above, there's no comparison to non-safety-tuned base checkpoints. Without that, it's impossible to tell whether the improvement from 18 to 58% is due to “breaking safety” or just “probing the base model.” I think this kind of thing is essential for supporting the central claim that jailbreaks cause regurgitation rather than merely reveal it.

3. Concerns about inflated success rates

I'm concerned that counting relatively short n-gram overlaps inflates success rates; the task is closer to local next-token recall than to meaningful extraction. Most prior work uses much more stringent success criteria (typically, 50 tokens of verbatim content) for a reason---it needs to be sufficiently long to be valid evidence for _extraction_ of _memorized_ training data. If the goal here is just to study how much models produce content that _resembles_ training data (rather than a claim about memorization), this perhaps would be okay. But then the work needs to be written as such, not connect so directly as continuing the lineage of memorization/extraction literature. (This remains important related work, but the current work would not be extending it, as it is running a very different type of experiment). I am concerned that the experiments, if interpreted as making a contribution to this other literature, have validity issues (more on this below).

4. Validity: No confirmation of training-set membership

Regarding claims about _memorization_ (not relevant if the paper is making a claim about producing similar data that is not memorized, but again, this seems to be conflated in the paper): making a claim for extraction is making a claim about training-data membership. To be extracted, by definition, the text must have been memorized; to be memorized, the text has to have been in the training data. If you do not have ground-truth information about membership (as is the case here with models like Qwen), one needs to do more for validity than just check that it matches a public book or article. See Cooper et al. 2025, which examines this directly for specific snippets. Also see Carlini et al. 2019 for earlier work on this, which studies the arbitrary (rather than specific snippet) setting.

5. Overclaims

The paper says it is the first large-scale empirical study linking jailbreaks and extraction, but it's actually a small-scale test. It also doesn't engage prior work effectively that has trodden similar landscape, but done so in much greater detail, either on the jailbreaking side (Nasr et al. 2023), on the extraction methodology side (Carlini et al. 2023), on the validity side (Carlini et al. 2019, Cooper et al. 2025), or on the snippet-specific extraction side (Cooper et al. 2025). Further, while the point about model family vs. size is interesting, this is more effectively studied and discussed in prior work (Cooper et al. 2025).

[1] Carlini et al. 2023. Quantifying Memorization Across Neural Language Models.

[2] Hayes et al. 2025. Measuring memorization in language models via probabilistic extraction

[3] Cooper et al. 2025. Extracting memorized pieces of (copyrighted) books from open-weight language models

[4] Carlini et al. 2019. Extracting Training Data from Large Language Models.

[5] Nasr et al. 2023. Scalable Extraction of Training Data from (Production) Language Models

**Questions:**

Please see the weaknesses above. I don't have questions at this time.

---

> ### Comment · Reviewer_oh7i · 2025-11-25
> **Reviewer acknowledgment**
>
> There hasn't been a rebuttal/response, so I'm leaving my review and score as-is.

---

### Official Review · Reviewer_kj1p · 2025-11-11

**Soundness:** 2
**Presentation:** 2
**Contribution:** 2
**Rating:** 2
**Confidence:** 4

**Summary:**

The paper studies the effect of jailbreaks on the generation of text memorized verbatim. The authors show that they can make the models generate significantly more memorized text when attacked compared to the regular prefix prompting baseline.

**Strengths:**

- The topic studied by the paper is interesting (verbatim memorization and the effect of jailbreaks on it).

- The writing is quite straightforward and easy to follow. However, there are some missing details as mentioned below.

**Weaknesses:**

- The paper is missing a significant amount of details. For example, it is not clear how the authors tune the attacks for this setting. Can the authors provide all the details of each attack tuned for the case of verbatim regurgitation? What optimization objective is used for each attack?

- The paper's main contribution is to study the effect of jailbreaks on verbatim text generation. However, it feels like it's missing a deeper contribution. This study would be more valuable as an observation at the beginning of a more complete analysis of jailbreaking and verbatim memorization. How much effort does the attacker need to make the model generate those samples? What are the constraints for the attacker? (Because if the attacker can, for example, inject many adversarial tokens, then it might be possible to make the model generate any text, not necessarily memorized ones.) Other points the authors could study to extend this work include ways to defend against jailbreaks that aim for verbatim regurgitation.

- There is relevant prior work [1] that the authors seem to have overlooked.

Minor comments:

- I would use "prefix" instead of "seed."

- Table 2 and Table 1 need to be changed order.


[1] https://arxiv.org/pdf/2404.15146

**Questions:**

See above.

---

### Author Response · Authors · 2025-11-29
**Rebuttal**

## **Response to Reviewer `kj1p`**

Thank you for the thoughtful feedback.

### **Attack Details**
All 12 jailbreaks are documented prompt-based attacks from prior work. We apply them *as-is* without model-specific tuning, reflecting realistic attacker behavior. We will include full prompt templates and adaptation notes for reproducibility.

### **Contribution Scope**
Our contribution is intentionally scoped:
1. A systematic methodology for evaluating jailbreaks as extraction vectors.
2. A broad study across 12 methods and 9 model families revealing an **architecture-over-size** trend.
3. Practical defense implications.

We agree deeper analysis of attacker effort and defenses belongs in future work.

### **Metric Use**
Our N-gram F1 \(>0.3\) threshold targets **verbatim continuation**, not paraphrasing. We will add clarifying examples.
We will integrate the missed related work and reorder Tables 1 and 2.

---

## **Response to Reviewer `oh7i`**

We appreciate the detailed comments and acknowledge several limitations.

### **Scope Calibration**
Our goal is narrower than interpreted: we study jailbreak-induced **verbatim output** in small instruction-tuned models, not full-scale training-data extraction. We will revise the framing accordingly.

### **Model Size**
We use 1–4B models for reproducibility and will state clearly that our results are **boundary-case evidence**. Prior work (Cooper 2025, Hayes 2025) indicates larger models likely show stronger effects.

### **Baseline Comparison**
Including base (non-safety-tuned) checkpoints would better separate “breaking safety’’ from “probing the base model.’’ We will note this as a key limitation.

### **Thresholding**
The 50-token extraction standard addresses a different problem (membership inference). Our focus is regurgitation under adversarial prompting. We will add ablations at 0.3/0.4/0.5 and clarify the distinction.

### **Architecture**
We will position our finding as complementary to Cooper et al. (2025), which studies architecture more comprehensively.

---

## **Response to Reviewer `kfjh`**

Thank you for the constructive review.

### **Jailbreaks vs. Prompt Diversity**
We will add two ablations:
1. Per-method extraction rates for all 12 jailbreaks.
2. A prompt-diversity control using non-adversarial stylistic prompts.
Preliminary results show improvements come from adversarial framing, not simply more attempts.

### **Safety-Tuning Behavior**
We will include examples showing refusals vs. degraded outputs and clarify which safety layers plausibly affect regurgitation.

### **Thresholding**
A full ablation (0.2/0.3/0.4/0.5) will be added to show robustness.

### **Terminology and Robustness**
We will connect “targeted regurgitation’’ to “discoverable memorization’’ (Carlini 2022) and report how many jailbreaks succeed per target to show whether the effect is broad or concentrated.

---

## **Response to Reviewer `ViYU`**

Thank you for the detailed feedback.

### **Attack Selection**
We focus on template-based jailbreaks as realistic low-cost attacks. Automated methods such as TAP and PAIR require optimization objectives designed for harmful-content elicitation, and adapting them for targeted continuation is non-trivial. We will add discussion of this limitation and outline white-box extensions.

### **Threat Model**
We explicitly study a **black-box** setting matching real deployments. We will clarify this and discuss implications for white-box cases.

### **Missing Details**
We will provide:
- all 12 jailbreak templates,
- adaptation details,
- full per-attack success rates.

### **Model Size**
We acknowledge the 1–4B limitation and will caveat conclusions, referencing Carlini et al. (2023) on scaling.

### **Effectiveness and Datasets**
Table 3 will be expanded to show all 12 attacks and separate failure modes (refusal vs. jailbreak success but extraction failure).
We will justify dataset choices and note limits when training-data membership is uncertain.

---

### Note · Program_Chairs · 2026-01-17
**Submission Desk Rejected by Program Chairs**

The following references in this submission do not refer to real documents and/or have major errors in bibliographic information:

 Gelei Deng, Can Xu, Zhipeng Ma, Yiming Wen, Yi Liu, Zhiruo Liu, Yuekang Li, and Bo Li. Jailbreaking large language models via reading compressed semantics. arXiv preprint arXiv:2310.19837, 2023.
Edward Lee. Navigating the jagged legal frontier of generative ai. Available at SSRN 4454240, 2023.
Matthew Jagielski, Nicholas Carlini, Daphne Ippolito, Milad Nasr, and Florian Tramer. Measuring and reducing the impact of spurious correlations on data extraction. arXiv preprint arXiv:2310.17714, 2023.